# Influence of reduced graphene oxide addition on kerf width in abrasive water jet machining of nanofiller added epoxy-glass fibre composite

**Kavimani V.**[1], **Gopal P. M.**[1], **Stalin B.**[2], **Balasubramani V.**[3], **Dhinakaran V.**[4], **Nagaprasad N.**[5], **Leta Tesfaye Jule**[6,7], **Krishnaraj Ramaswamy**[6,8]*

**1** Department of Mechanical Engineering, Centre for Material Science, Karpagam Academy of Higher Education, Coimbatore, Tamil Nadu, India, **2** Department of Mechanical Engineering, Anna University, Madurai, Tamil Nadu, India, **3** Department of Mechanical Engineering, Thiagarajar College of Engineering, Madurai, Tamil Nadu, India, **4** Department of Mechanical Engineering, Chennai Institute of Technology, Kundrathur, Chennai, Tamil Nadu, India, **5** Department of Mechanical Engineering, ULTRA College of Engineering and Technology, Madurai, Tamil Nadu, India, **6** Centre for Excellence-Indigenous Knowledge, Innovative Technology Transfer and Entrepreneurship, DambiDollo University, Addis Ababa, Ethiopia, **7** Department of Physics, DambiDollo University, Addis Ababa, Ethiopia, **8** Department of Mechanical Engineering, DambiDollo University, Addis Ababa, Ethiopia

* prof.dr.krishnaraj@dadu.edu.et

**Data Availability Statement:** All relevant data are within the paper and its Supporting Information file.

## Abstract

The present study aims to develop a novel hybrid polymer composite with reduced graphene oxide (rGO) as filler and optimize its Abrasive Water Jet Machining (AWJM) parameters for reduced kerf width. The influence of rGO addition on kerf width is analysed in detail along with Pump pressure (bar), Transverse speed (mm/min) and Standoff distance(mm). The experiments are designed based on Taguchi's orthogonal array techniques in which L27 is adopted for three input parameters at three levels. The influence of each factor is used to identify the significance of selected parameters over kerf width, and it was found that stand of distance has a major effect over kerf width irrespective of rGO %. The addition of rGO filler has a significant effect on kerf width, which decreases with the addition of rGO up to 0.2% and kerf width increases for further addition of rGO.

## Introduction

Polymer matrix composite has a major role in the industrial sector owing to its unique merits such as high stiffness, lower in weight while compared with traditional metals, and capability to mould complex shaped products with better damping behaviour. Nowadays, carbonaceous nanomaterials, nano clays and metal oxides such as graphene, CNT, BN, $SiO_2$, $TiO_2$, Montmorillonite etc., are incorporated into the polymer matrix to improvise its basic and functional properties [1–4]. Herein graphene-based nanofillers are incorporated in an epoxy matrix and attain notable improvement in mechanical properties that includes flexural strength, tensile strength etc.; also, some studies reveal that usage of carbonaceous material made the developed composite suitable for fabricating the components of electronic devices [5–8].

**Funding:** The authors received no specific funding for this work.

**Competing interests:** The authors have declared that no competing interests exist.

Generally, continuous fibers are utilized in developing polymer matrix composite for structural applications, especially in aerospace and transport industries. Due to the usage of continuous fiber, slots and hole making over fibre-reinforced composites is a difficult task to date [9–13]. Machining fibre reinforced composite using conventional technique results in poor dimensional accuracy, fiber pull-out, high operating cost, fibre damage, higher tool wear rate, interlaminar failure, surface damage etc. To overcome these facts, unconventional machining process such as ultrasonic machining, abrasive water jet machining, and water jet machining is adopted for machining fibre-reinforced composite [14,15]. Among them, Abrasive water jet machining was preferred in industries owing to its unique characteristics like better cutting speed, low-temperature operation and nil dust generation during machining of composite.

In AWJM, the surface of the composite is not affected by thermal stress damages due to the action of water coolant, and the presence of abrasives particles improves the material removal process. In general, silicon carbide, garnet, aluminium oxide etc., are generally used as abrasives in AWJM [16,17]. The abrasive water jet machining performance is mostly influenced by process parameters, quantity and type of filler and reinforcement addition in polymer matrix composite. Hence, it is inevitable to select proper control factors to attain better surface quality and machinability in fibre reinforced composite machining [18,19]. For selecting optimal machining parameters, statistical tools like Taguchi, Genetic algorithm, ANN, Grey relation analysis, Response surface methodology etc., are used by researchers [20–22]. There are few researches that deal with abrasive water jet machining of fibre-reinforced polymer composite. Kalyan Kumar Singh and Raju Kumar Thakur developed epoxy-glass fibre composite with the addition of various wt.% of carbon nanotube fillers and optimized its AWJM characteristics [23]. Process parameter like Traverse rate, jet pressure, and standoff distance was selected to understand their effects over kerf taper, surface roughness and material removal rate. Results reveals that increase in level of Traverse rate improvise the output parameter and also depicts that filler materials addition is a major influencing factor for kerf taper. Grey relation analysis was utilized by G. Anand and co-workers to attain optimal parameter for AWJM of polymer matrix composite [24]. Results depict that abrasive flow rate act as significant parameter on influencing output parameters. K.R. Sumesh and co-worker utilized AWJM to understand the machinability behaviour of fillers reinforced polymer composite. Process parameter such as Water jet pressure, Traverse speed, Standoff distance was selected as input parameter; kerf angle and surface roughness as output response. Results reveals that filler weight percentage has major influence over output parameter, also observed that lower range of Traverse speed and Standoff distance decreases the kerf angle and surface roughness of developed composite [25]. Arunkumar et al. used response surface methodology to investigate the machinability of graphene filler reinforced polymer composite using AWJM. Surface roughness was selected as output parameter whereas Water Pressure, Traverse speed and Standoff distance are selected as input parameters. Results reveals that better surface finishing for graphene filler-based composite can be attained by increasing water pressure and decreasing the nozzle transfer speed [26].

Based on the reviewed literature, it can be depicted that the addition of graphene-based hybrid fillers increases the functional behaviour of polymer matrix composite. However, there are only a few works of literature based on the machinability behaviour of hybrid graphene filler reinforced polymer matrix composite. Based on this fact, an attempt has been made to understand its machinability behaviour by adopting an abrasive water jet machining process. The compression moulding process has been used to develop reduced graphene- Mont Morillonite Nano clay filler reinforced epoxy-glass fiber composite. The AWJM experimental design is done through the Taguchi method, and the effect of reduced graphene oxide filler is also investigated.

## Materials and method

In this research, LY 556 grade epoxy along with HY951 grade hardener is selected as matrix material by maintaining the ratio of 1:10. The above-mentioned epoxy grade exhibits an acceptable range of dimension stability which makes it a widely used base matrix. By the addition of an optimal percentage of reinforcement and filler material, the basic and functional properties of the epoxy might be improved. 200 GSM glass fibres are selected as reinforcement fibre, and it is added with epoxy matrix at 30 wt. %. Based on available literature surveys, it was found that the optimal level of glass fibre is 30 wt. % to improvise the structural strength of the epoxy matrix. Herein reduced graphene oxide and Mont Morillonite Nano clay (MMT) with the size range of 50–100 nm was used as filler material in order to improve the thermal stability and binding efficiency among matrix material with reinforcement. A detailed procedure for reduced graphene synthesis was already reported in our previous studies, and MMT was purchased from Ad-nano Tech. In this research weight percentage of MMT was fixed as 1.5, and the wt. % of reduced graphene oxide is varied at a constant interval of 0.1 up to 0.3.

### Composite fabrication

The compression moulding approach is adopted to develop the proposed combinations of epoxy–glass fibre composites. As per the research plan, calculated wt.% of filler material is ultrasonicated separately for 1hr with an organic solvent. After attaining proper dispersion, the filler solution is mixed together for 2 hr through ultra-sonication. The dispersed hybrid filler is added to the epoxy matrix and then mechanically stirred for 2 hr at a speed of 1200 rpm. Further HY951 hardener is added to the matrix filler mixture with a calculated ratio of 1: 10. Initially, the hand layup method is used to coat the matrix filler mixture over glass fiber. The stacking of coated glass fibre is made with a plate thickness of 30 mm in dimension, followed by degassing. The stacked polymer matrix is pressed using a hot press under 15 MPa pressure, followed by a curing process [4]. The same procedure is followed to develop the remaining set of composites with different amounts of filler addition. The developed composite with a varying weight percentage of r-GO is depicted in Fig 1. This developed composite exhibits better mechanical and flame retardancy properties up to 0.3 wt. % addition of r-GO fillers. The detailed investigation of the mechanical behaviour of the developed composite was illustrated in our previous report [27].

### Experimental design

The machinability studies of the fabricated composite are carried out using Abrasives Water Jet Machine (AWJM). Herein the utilized AWJM consists of a high-pressure pump (DIPS6-2230) combined with a water jet cutter (DWJ1313-FB) with a 90˚ impact angle and 0.7 mm diameter orifice. Fig 2A depicts the visualization of AWJM with garnet sand as the abrasive particle (80 mesh). Based on the literature survey and experts console, stand-off distance, transverse speed and pump pressure are selected as input parameters. The traditional Taguchi method is adopted for experimental planning and to understand the effects of the control parameter. In this research 3 factors, each at 3 levels, are selected for the experimental plan (Table 1). Kerf width is taken as the output parameter, and it is measured by calculating the distance between machined slots using a Leica microscope coupled with an image analyser, as depicted in Fig 2B. Three reading has been taken from different places of slots between machined surfaces for the exactness of the reading, and the mean values are taken.

In order to minimize the experimental cost and time, the L27 orthogonal array is chosen by using Statistical Software R programming. Effects of input parameter over output response have been understood by Taguchi's method. In this research, kerf width is selected as the

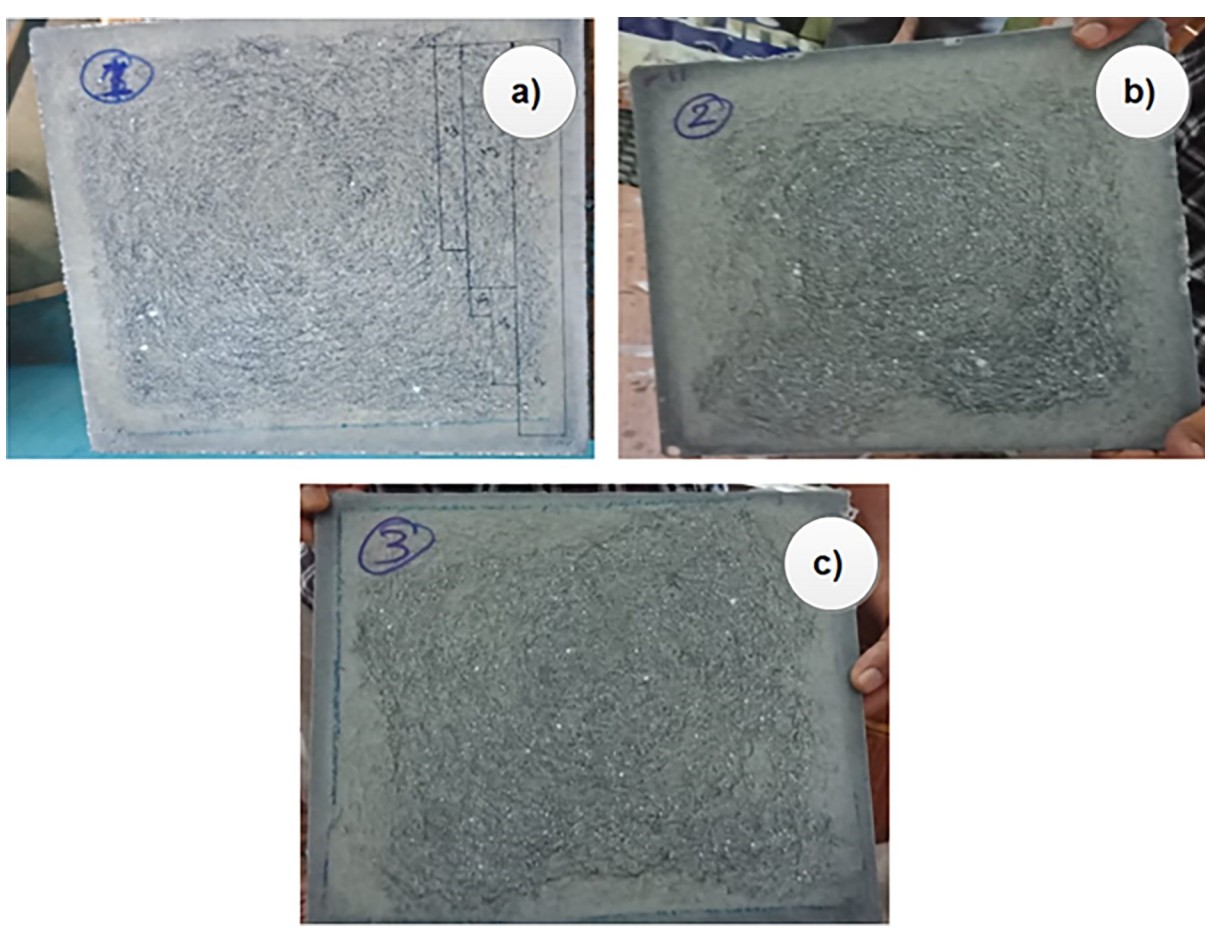

**Fig 1.** Composite plate with varying r-GO addition a) 0.1 wt.% b)0.2 wt.% c) 0.3wt.%.

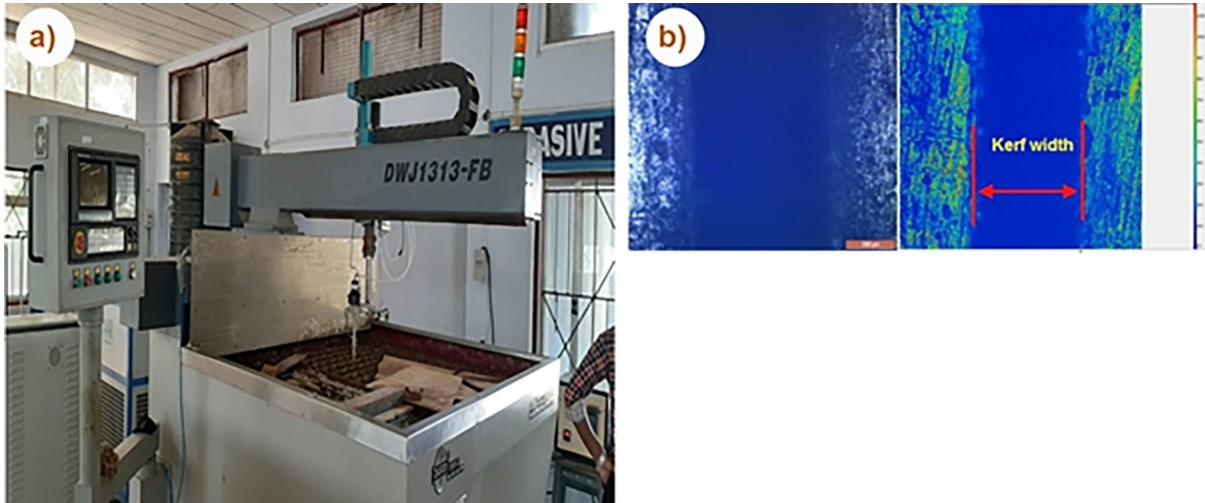

**Fig 2.** a) Abrasive water jet machine b) optical image of machine surface for kerf width measurement.

**Table 1. Process parameter for machinability studies.**

| Factors | Notation | Level 1 | Level 2 | Level 3 |
|---|---|---|---|---|
| **Pump pressure (bar)** | A | 220 | 240 | 260 |
| **Transverse speed (mm/min)** | B | 30 | 40 | 50 |
| **Stand of distance (mm)** | C | 1 | 2 | 3 |

output response and the smaller, the better criteria have been chosen since minimal kerf width will improve the dimensional accuracy. Figs 3 and 4 display the L27 orthogonal array data in matrix scatter plot and hierarchical clustering among input and output variables using R programming. Scatter-plot matrix is used to identify the correlation among the set of variable pairs. These pairwise correlations can be organised in the form of a matrix. In general present matrix pair in diagonal order depicts a better correlation among the matrix pairs and absents of outliers in the attained output data. A hierarchical cluster is an analysis worked by the algorithm in which similar data are grouped in the form of clusters. Herein each cluster is varied from each other, and the values are widely similar to each other. From Fig 4, it can be noted that the values of the output parameter are grouped into four clusters with two expressive colour blue and brown (dark and lighter). The L27 orthogonal array of developed composite is listed in Table 2. These are ordered at the range of -1 to 1 with respect to the input parameter. The values of output parameters are predominant ordering from top to bottom displayed in the form of lighter and dark colours of blue and brown.

## Result and discussion

The kerf width values measured with the aid of an optical microscope for the polymer composite specimen with and without filler addition are given in Fig 5. These values are converted into a signal to noise ratio to find the influence of each factor over kerf width.

### Effect of process parameter on epoxy glass fibre composite

The response Table 3 shows the mean signal to noise ratio values calculated based on the experimental results for composite without r-GO addition. It is clearly evident from the table that the pump pressure has a negative impact over kerf width. Kerf width increased considerably with an increase in pump pressure owing to the higher cutting force acting over the developed specimen. When the pump pressure is high, the kinetic energy of the jet is also high, which opens a wider slot over the work surface, i.e., kerf width increases. On the other hand, the increase in transverse speed results in reduced kerf width. But the kerf width increases greatly with an increase in standoff distance. Jet divergence is considered the main reason behind the increased kerf width at a higher standoff distance. It can also be notified from Table 3 that stand of distance acts as the major significant parameter for Kerf width. This fact may be owing to the expansion and disintegration of the abrasive water jet particle through boundary interactions.

The combined effect of process parameters on over-developed composite materials is shown in Fig 6A–6C. It can be noted that the kerf width is higher at maximum values of pump pressure and standoff distance (Fig 6A). Higher level of standoff distance and pump pressure results in discharging of water jet particles at high-pressure conditions, followed by lower surface contact between specimen and nozzle. At this condition, water jet particles impact over the specimen surface at higher pressure, which results in incremental kerf width. Likewise, increases in pump pressure and transverse pressure results in higher kerf with values (Fig 6B). This might be due to the combined effect of process parameters that stimulate the discharging

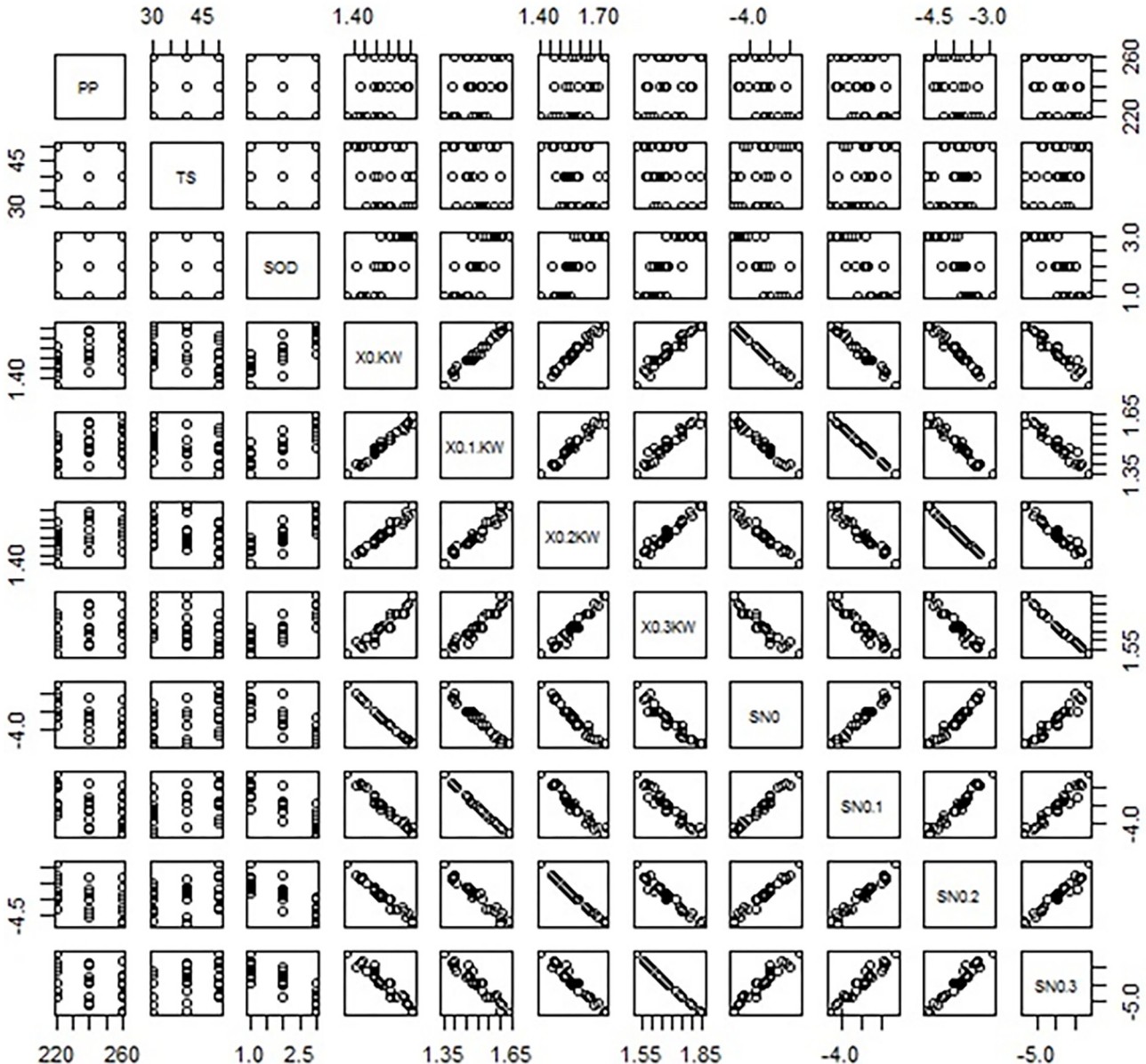

**Fig 3. Matrix scatterplot of L27 OA dataset.**

of abrasive particles under high velocity and pressure [26]. The impact of these particles over the composite surface at the above said condition results in increased kerf width. Fig 6C depicts that kerf width values are minimal at lower standoff distance and transverse speed; during this process condition, the distance between the composite surface and nozzle is lower with a minimum velocity of abrasive water jet particles. This fact might be the reason for the lower kerf width at the combined effect. Based on ANOVA results, it can be noted that the selected parameters have a considerable significant effect on kerf width (Table 4). As the p-value for all the input parameters is less than 0.05, it is said to be significant. Also, the R-Square value of 95.62% shows the reliability of the analysis performed. The contribution percentage chart given in Fig 7 confirms that the stand of distance (63.33%) has a major effect over kerf width, followed by pump pressure (21.91%) and transverse speed (14.77%).

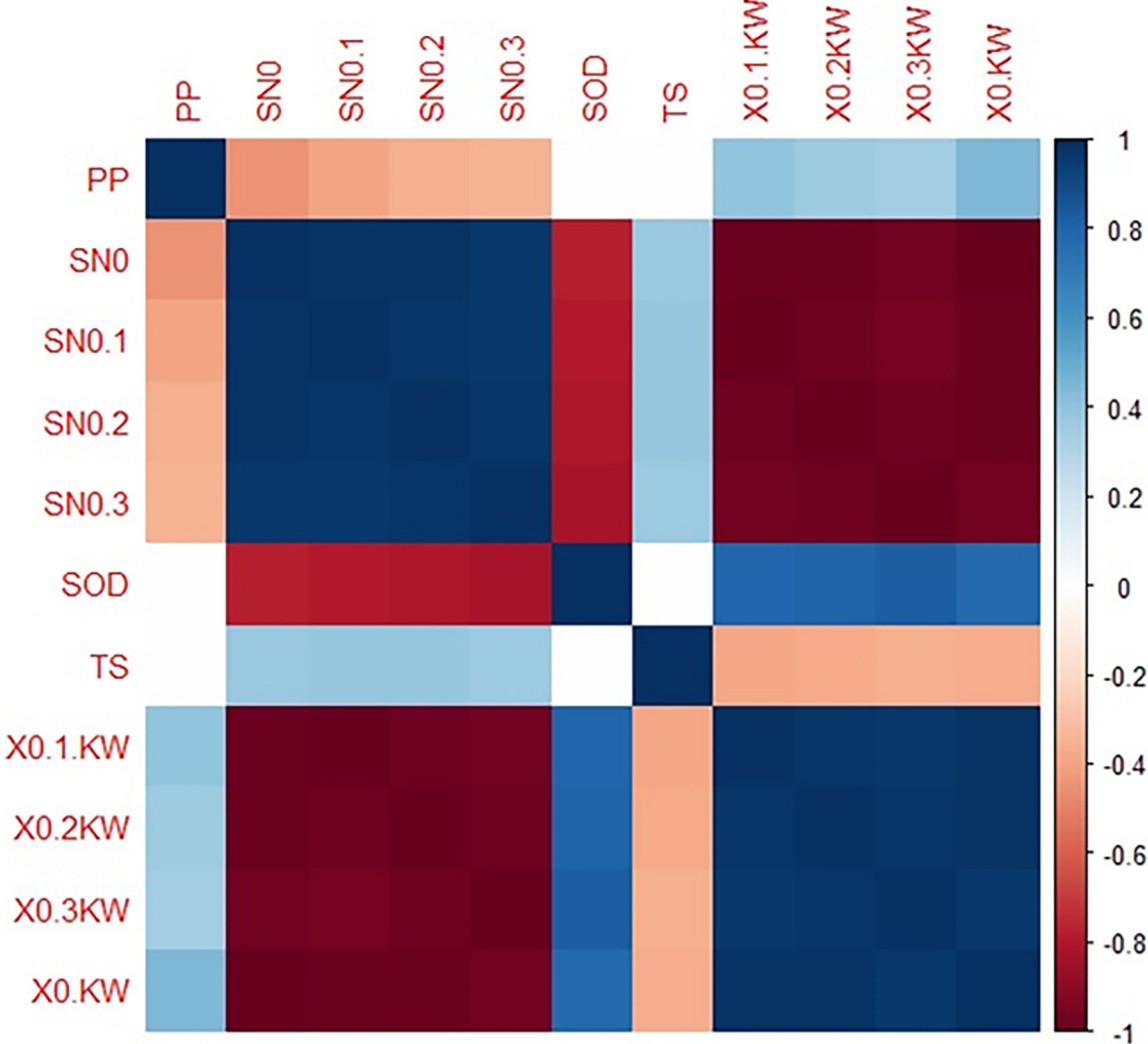

**Fig 4. Hierarchical clustering of correlation coefficient matrix of input variables.**

## Effect of process parameters for 0.1 wt. % of filler loaded composite

From Fig 5, it can be observed that kerf width values are minimal for reduced graphene oxide filler reinforced composites. These filler particles have better efficiency in improvising the interfacial interaction among that matrix and reinforcement that improve the load-carrying capability of matrix materials. Due to this strengthening mechanism, the developed composite can withstand the impact of high-velocity abrasive particles and decreases the surface damage, consequently minimal kerf width. Another reason might be the higher surface area of filler material that decreases the effect of high-velocity abrasive particles during machining conditions [28]. Herein the standoff distance act as a significant parameter for the composite loaded with 0.1 wt.% reduced graphene oxide (Table 5). A higher level in standoff distance values results in more expansion of jet that increases the kinetic energy losses in abrasive water jet particles; thus, kerf width increases. It can also be noted that transverse speed has a minimal effect on kerf width for reduced graphene oxide loaded composite.

**Table 2. L27 orthogonal array of the developed composite.**

| A | B | C | Kerf width (µm) | | | | S/N ratio | | | |
|---|---|---|---|---|---|---|---|---|---|---|
| | | | 0% | 0.1% | 0.2% | 0.3% | 0% | 0.1% | 0.2% | 0.3% |
| 220 | 30 | 1 | 1.46 | 1.41 | 1.50 | 1.61 | -3.29 | -2.98 | -3.52 | -4.14 |
| 220 | 30 | 2 | 1.51 | 1.49 | 1.56 | 1.67 | -3.58 | -3.46 | -3.86 | -4.45 |
| 220 | 30 | 3 | 1.56 | 1.54 | 1.64 | 1.75 | -3.86 | -3.75 | -4.30 | -4.86 |
| 220 | 40 | 1 | 1.43 | 1.39 | 1.48 | 1.57 | -3.11 | -2.86 | -3.41 | -3.92 |
| 220 | 40 | 2 | 1.49 | 1.48 | 1.54 | 1.61 | -3.46 | -3.41 | -3.75 | -4.14 |
| 220 | 40 | 3 | 1.56 | 1.52 | 1.59 | 1.72 | -3.86 | -3.64 | -4.03 | -4.71 |
| 220 | 50 | 1 | 1.37 | 1.35 | 1.40 | 1.52 | -2.73 | -2.61 | -2.92 | -3.64 |
| 220 | 50 | 2 | 1.41 | 1.40 | 1.46 | 1.59 | -2.98 | -2.92 | -3.29 | -4.03 |
| 220 | 50 | 3 | 1.52 | 1.48 | 1.57 | 1.67 | -3.64 | -3.41 | -3.92 | -4.45 |
| 240 | 30 | 1 | 1.52 | 1.51 | 1.55 | 1.67 | -3.64 | -3.58 | -3.81 | -4.45 |
| 240 | 30 | 2 | 1.55 | 1.52 | 1.59 | 1.67 | -3.81 | -3.64 | -4.03 | -4.45 |
| 240 | 30 | 3 | 1.64 | 1.61 | 1.67 | 1.80 | -4.30 | -4.14 | -4.45 | -5.11 |
| 240 | 40 | 1 | 1.49 | 1.46 | 1.53 | 1.58 | -3.46 | -3.29 | -3.69 | -3.97 |
| 240 | 40 | 2 | 1.51 | 1.48 | 1.55 | 1.66 | -3.58 | -3.41 | -3.81 | -4.40 |
| 240 | 40 | 3 | 1.63 | 1.60 | 1.69 | 1.79 | -4.24 | -4.08 | -4.56 | -5.06 |
| 240 | 50 | 1 | 1.43 | 1.39 | 1.47 | 1.57 | -3.11 | -2.86 | -3.35 | -3.92 |
| 240 | 50 | 2 | 1.49 | 1.47 | 1.53 | 1.64 | -3.46 | -3.35 | -3.69 | -4.30 |
| 240 | 50 | 3 | 1.59 | 1.56 | 1.64 | 1.75 | -4.03 | -3.86 | -4.30 | -4.86 |
| 260 | 30 | 1 | 1.49 | 1.46 | 1.52 | 1.63 | -3.46 | -3.29 | -3.64 | -4.24 |
| 260 | 30 | 2 | 1.62 | 1.57 | 1.65 | 1.75 | -4.19 | -3.92 | -4.35 | -4.86 |
| 260 | 30 | 3 | 1.66 | 1.64 | 1.72 | 1.85 | -4.40 | -4.30 | -4.71 | -5.34 |
| 260 | 40 | 1 | 1.49 | 1.45 | 1.52 | 1.64 | -3.46 | -3.23 | -3.64 | -4.30 |
| 260 | 40 | 2 | 1.56 | 1.52 | 1.58 | 1.67 | -3.86 | -3.64 | -3.97 | -4.45 |
| 260 | 40 | 3 | 1.66 | 1.60 | 1.73 | 1.84 | -4.40 | -4.08 | -4.76 | -5.30 |
| 260 | 50 | 1 | 1.44 | 1.40 | 1.47 | 1.56 | -3.17 | -2.92 | -3.35 | -3.86 |
| 260 | 50 | 2 | 1.53 | 1.49 | 1.56 | 1.68 | -3.69 | -3.46 | -3.86 | -4.51 |
| 260 | 50 | 3 | 1.61 | 1.58 | 1.63 | 1.74 | -4.14 | -3.97 | -4.24 | -4.81 |

Table 6 depicts the ANOVA results for 0.1 wt.% reduced graphene oxide loaded composite; it confirms that the selected parameters have significant effects over kerf width. As the p-value for all the input parameters is less than 0.05, it is said to be significant. Also, the R-Square value of 95.48% shows the reliability of the executed analysis. From ANOVA results, the contribution percentage of each selected parameter can be calculated based on sequential sums of square values. It can be depicted (Fig 8) that the stand of distance (65.99%) has a major effect over kerf width, followed by pump pressure (18.57%) and transverse speed (15.44%). When compared with the results of the composite without filler, the effect of pump pressure over kerf width was slightly reduced for 0.1% filler added PMC.

The combined effect of machining parameters over Kerf width of 0.1 wt. % of reduced graphene oxide loaded composite is depicted in Fig 9A–9C. Higher values of transverse speed at minimal pump pressure exhibit lower kerf width values. Herein addition of reduced graphene oxide filler results in higher impact resistance of the developed composite, and this fact increases the energy absorbing capacity of the matrix material. This might be the reason for variation or minimal kerf values for reduced graphene oxide loaded composite while compared with composite developed without the addition of filler materials. Likewise, comparatively, the significance of the machining parameter is the same for the non-filler and filler loaded composites; however, the values of kerf width are minimal for filler loaded composites

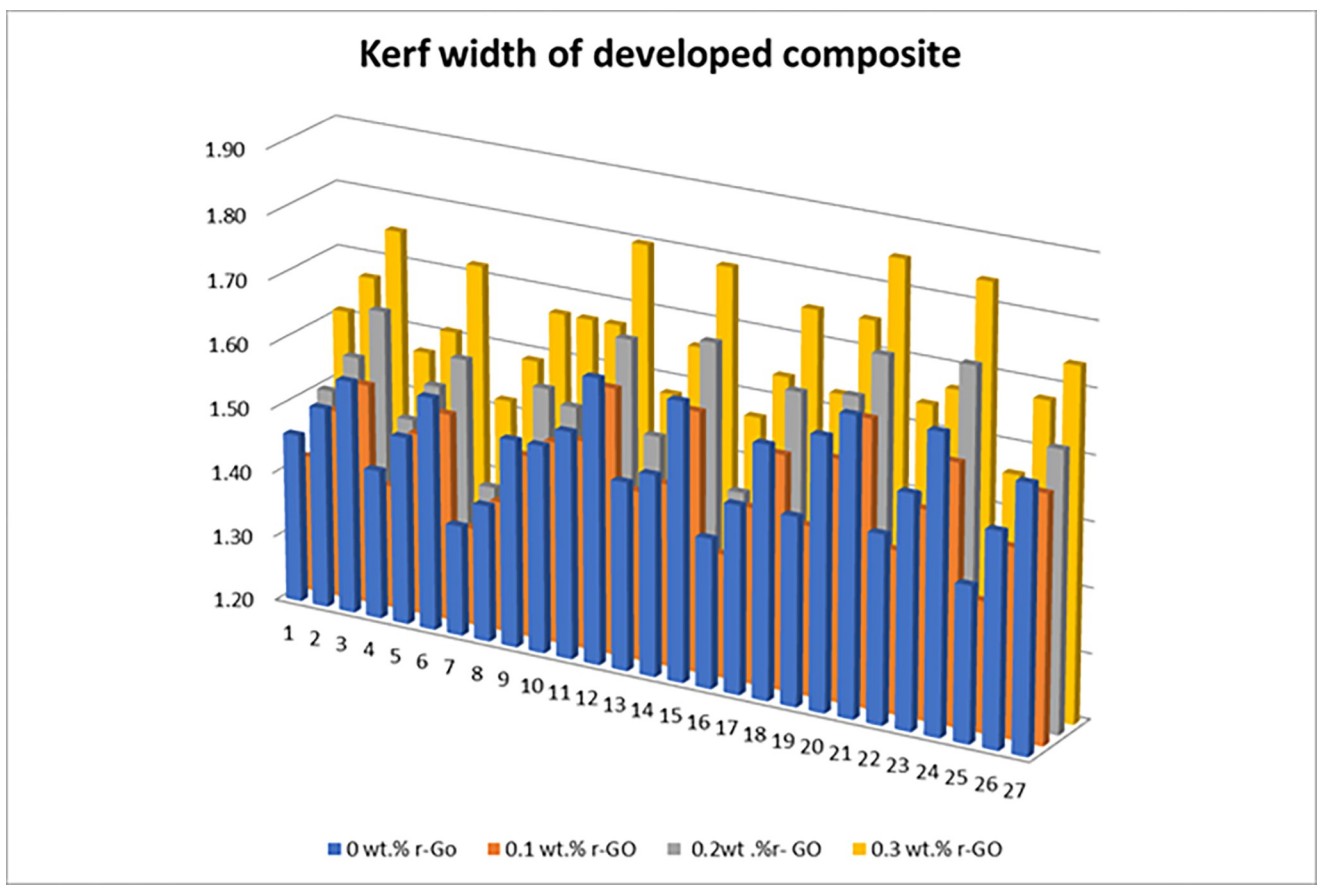

**Fig 5. Effect of machining parameter on Kerf width of developed composite.**

due to the above said facts. It can also be noted that increment in values of pump pressure results in higher kerf width owing to increased kinetic energy that imposing over the composite surface, which widens the narrow slot [29]. Likewise, incremental values of kerf width at higher traverse speed might be due to the quicker passing of high-velocity abrasive particles that allows the minimal quantity of abrasive particles strikes over the targeted areas, thus decreasing the kerf width.

### Effect of process parameters for 0.2 wt. % of filler loaded composite

The kerf width values of 0.2 wt.% reduced graphene oxide loaded composites are converted into SN ratio, and its mean values are depicted in Table 7. It can be observed that kerf width values are higher for 0.2 wt.% reduced graphene oxide loaded composite when compared with

**Table 3. Response table for glass fibre epoxy composite.**

| Level | Pump Pressure (bar) | Transverse speed (mm/min) | Stand of Distance (mm) |
|-------|---------------------|---------------------------|------------------------|
| 1 | 1.479 | 1.557 | 1.458 |
| 2 | 1.539 | 1.536 | 1.519 |
| 3 | 1.562 | 1.488 | 1.603 |
| Delta | 0.083 | 0.069 | 0.146 |
| Rank | 2 | 3 | 1 |

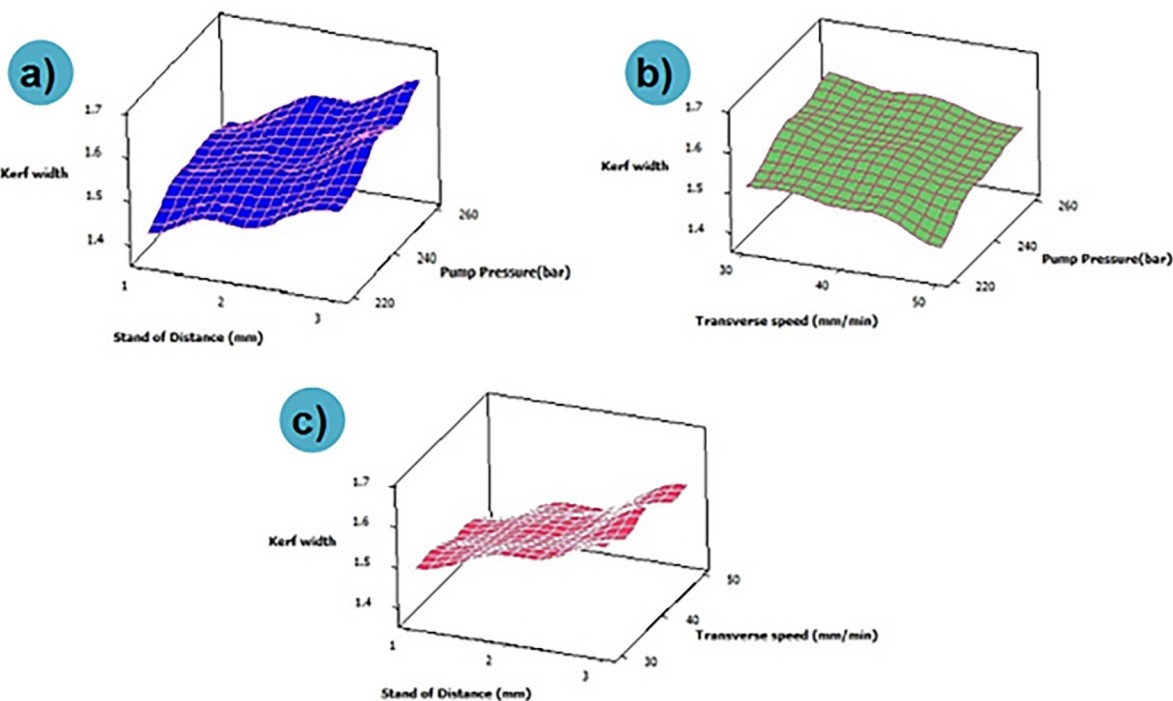

**Fig 6. Influence of AWJM parameters on kerf width for PMC without filler.**

0.1 wt.% reduced graphene oxide composite. This fact might be due to the clustering effect of nanofiller that results in the formation of voids that weaker the interfacial bonding among the matrix and glass fibre. Likewise, the presence of 1.5 wt.% MMT particles and 0.1 wt.% of reduced graphene oxide might be the suitable loading percentage for the proposed composite. Further variation in reduced graphene oxide loading might decrease the interaction among the fibre and matrix. From Table 7, it can be depicted that standoff distance act as the most significant parameter over kerf width where an increase in standoff distance increases the divergence of abrasive water jet particles; hence the kerf width increases. During the above-said machining condition, water jet particles travel a long way before impact over the composite surface resulting in the formation of jet divergence. This fact increases the diameter of the jet, and an increment in jet diameter increases the width of the cutting area, thus increasing the kerf width. Pump pressure has lower significance over kerf width; herein, an increment in the range of pump pressure decreases the kerf width. Since the increase in pump pressure increases the kinetic energy of abrasive water particles that form a turbulence flow of jet with higher pressure, it increases the kerf width.

**Table 4. ANOVA for glass fibre epoxy composite.**

| Source | DF | Seq SS | Adj SS | Adj MS | F—Test | P—value |
|---|---|---|---|---|---|---|
| Pump pressure (bar) | 2 | 0.033267 | 0.033267 | 0.016633 | 47.83 | 0 |
| Transverse speed (mm/min) | 2 | 0.022422 | 0.022422 | 0.011211 | 32.24 | 0 |
| Stand of Distance (mm) | 2 | 0.096156 | 0.096156 | 0.048078 | 138.24 | 0 |
| Error | 20 | 0.006956 | 0.006956 | 0.000348 | | |
| Total | 26 | 0.1588 | | | | |

R-Square value = 95.62%.

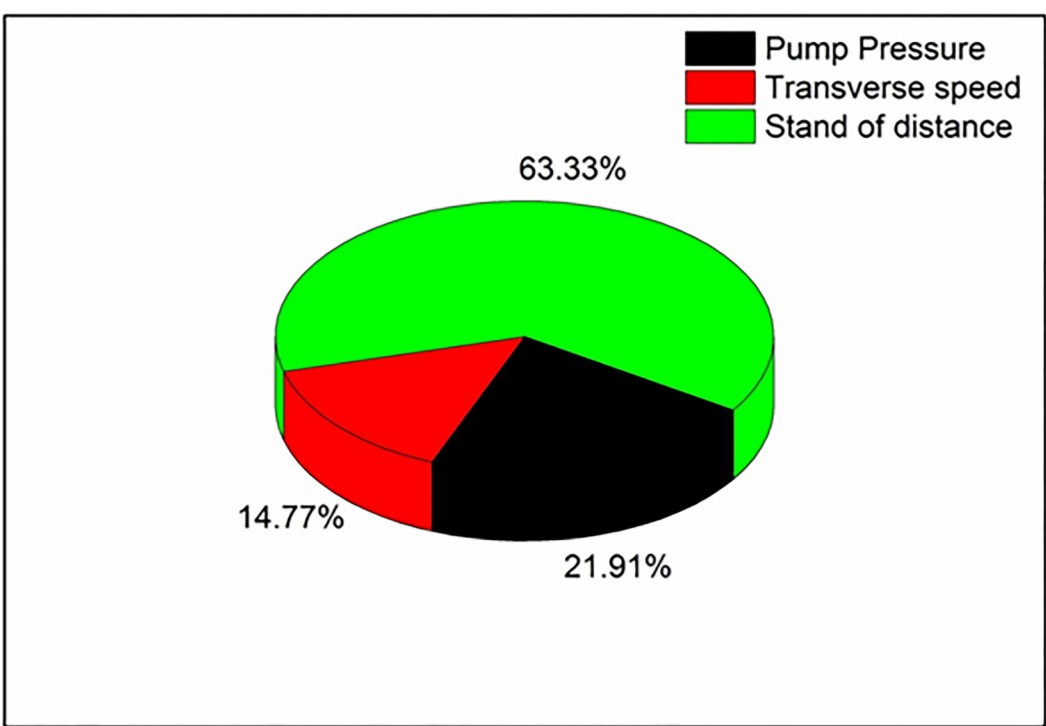

**Fig 7. Contribution of machining parameters in epoxy glass fibre composite.**

ANOVA table for the composite with 0.2% filler confirms that the parameters considered are having a significant effect over kerf width (Table 8). As the p-value for all the input parameters is less than 0.05, it is said to be significant. Also, the R-Square value of 94.69% shows the reliability of the analysis performed. The contribution percentage confirms that the stand of distance (69.52%) has a major effect over kerf width, followed by pump pressure (14.69%) and transverse speed (15.79%). The effect of pump pressure over kerf width further reduced to 14.69% for 0.2% filler added PMC from 18.57% for PMC with 0.1% filler and 21.91% for PMC without filler(Fig 10).

From Fig 11 it can be noted that kerf width is minimal for higher transverse speed and lower standoff distance. Likewise, minimal kerfwidth was depicted for the combination of lower pump pressure and higher transverse speed. During the above-said machining condition, the abrasive jet particles will pass through the hybrid filler which has a better surface area and high energy absorbing capability. Due to this fact, the abrasive particles might drop or loses their kinetic energy, and further impact over the composite surface results in loss of sharpness; only a few particles might be able to infiltrate the composite surface, and another

**Table 5. Response table of 0.1 wt. % filler loaded composite.**

| Level | Pump Pressure (bar) | Transverse speed (mm/min) | Stand of Distance (mm) |
|---|---|---|---|
| 1 | 1.451 | 1.528 | 1.424 |
| 2 | 1.511 | 1.5 | 1.491 |
| 3 | 1.523 | 1.458 | 1.57 |
| Delta | 0.072 | 0.07 | 0.146 |
| Rank | 2 | 3 | 1 |

**Table 6. ANOVA of 0.1 wt. % filler loaded composite.**

| Source | DF | Seq SS | Adj SS | Adj MS | F—Test | P—value |
|---|---|---|---|---|---|---|
| **Pump pressure(bar)** | 2 | 0.026896 | 0.026896 | 0.013448 | 39.25 | 0 |
| **Transverse speed (mm/min)** | 2 | 0.022363 | 0.022363 | 0.011181 | 32.64 | 0 |
| **Stand of Distance (mm)** | 2 | 0.095563 | 0.095563 | 0.047781 | 139.47 | 0 |
| **Error** | 20 | 0.006852 | 0.006852 | 0.000343 | | |
| **Total** | 26 | 0.151674 | | | | |

R-Square value = 95.48%.

abrasive particle may retard over the composite surface. It can also be visualized that kerf width is maximum at a higher level of stand-off distance. An increase in standoff distance results in disintegration and expansion of water jet further, thus increasing the kerf width. A higher value of pump pressure and lower values of transverse speed increases the kerfwidth. This fact may be an effect of selecting material process as well as the various ratios on the kinetic energy of the jet used in machining the specimen.

## Effect of process parameters for 0.3 wt. % of filler loaded composite

From Fig 5, it can be depicted that kerf width values are higher for composite developed with 0.3 wt.% reduced graphene oxide loading. This might be due to weaker interaction among the selected hybrid fillers. It can also be depicted that a combination of lower wt.% of reduced graphene oxide exhibits lower kerf width. Likewise, Kerf width increases with amplification in pump pressure and stand-off distance, whereas Kerf width decreases with an increase in

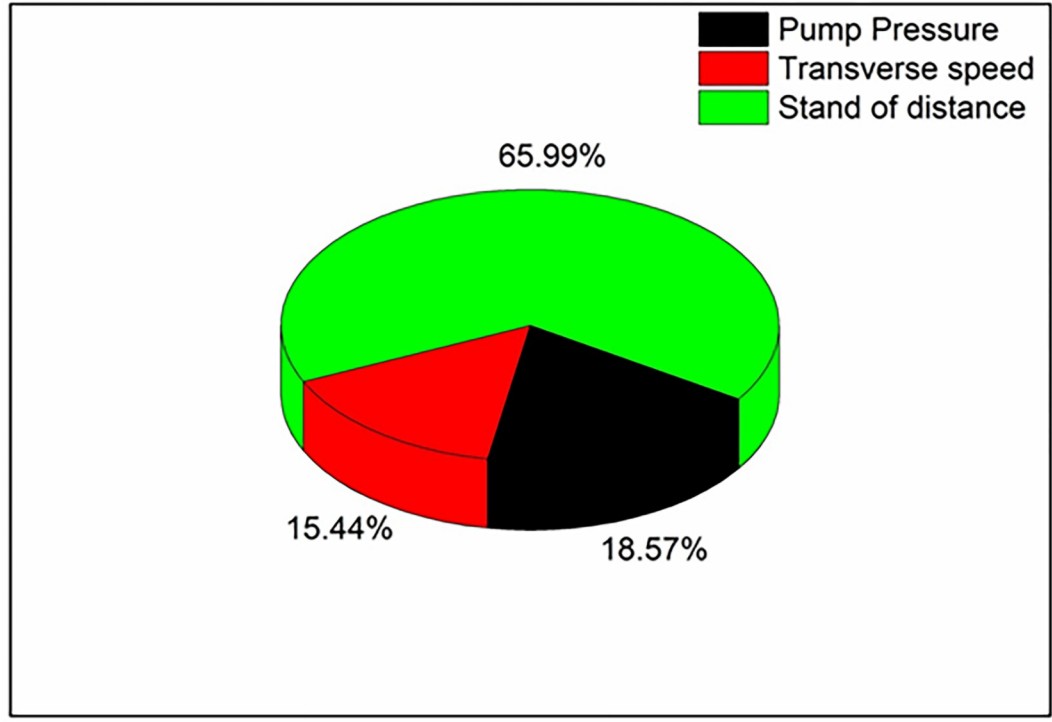

**Fig 8. Contribution of machining parameters in 0.1 wt. % filler loaded composite.**

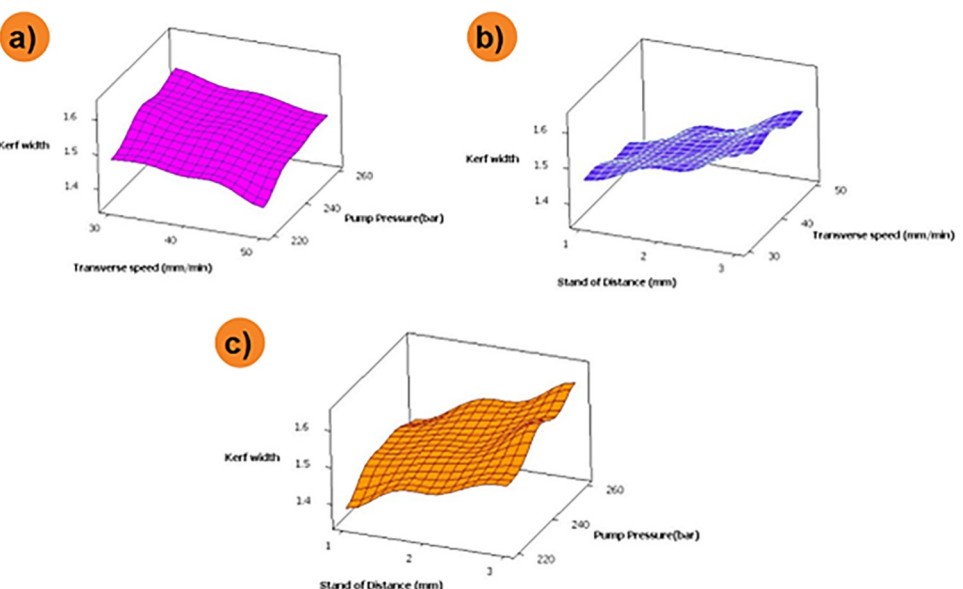

**Fig 9. Influence of AWJM parameters on kerf width for 0.1 wt.** % of filler loaded composite.

transverse speed. Table 9 showcases that pump pressure has a lower influence on over-developed composite; it might be due to increased loading of reduced graphene oxide filler. The addition of reduced graphene oxide filler increases the impact strength, due to which the developed composite gains the ability to withstand the load applied by abrasive particles. This might be the reason for the lesser significance of pump pressure in a fabricated composite. Herein, transverse speed acts as the second parameter to influence kerf width. At lower transverse speed, a greater number of abrasive waster jet particles are involved in the machining process that reducing the chance of interfacial rebounding of abrasive over the composite surface, thus increasing the kerfwidth [4].

Table 10 showcases the ANOVA results of the composite with 0.3% reduced graphene oxide loading. It confirms that the parameters considered are having a significant effect over kerf width. As the p-value for all the input parameters is less than 0.05, it is said to be significant. Also, the R-Square value of 94.50% shows the reliability of the analysis performed. The contribution percentage chart given in Fig 12 confirms that the stand of distance (73.44%) has a major effect over kerf width, followed by pump pressure (12.86%) and transverse speed (13.7%). The effect of pump pressure over kerf width further reduced to 12.86% for 0.3% filler added PMC from 14.69% for PMC with 0.2% filler, 18.57% for PMC with 0.1% filler and 21.91% for PMC without filler.

Fig 13 depicts the combinational effect of machining parameters for reduced graphene oxide loaded composites. It can be noted that increasing the order of pump pressure and

**Table 7. Response table of 0.2 wt. % reduced graphene oxide loaded composite.**

| Level | Pump Pressure (bar) | Transverse speed (mm/min) | Stand of Distance (mm) |
|---|---|---|---|
| 1 | 1.527 | 1.6 | 1.493 |
| 2 | 1.58 | 1.579 | 1.558 |
| 3 | 1.598 | 1.526 | 1.653 |
| Delta | 0.071 | 0.074 | 0.16 |
| Rank | 3 | 2 | 1 |

**Table 8. ANOVA table for 0.2 wt. % filler loaded composite.**

| Source | DF | Seq SS | Adj SS | Adj MS | F—Test | P—value |
|---|---|---|---|---|---|---|
| Pump pressure(bar) | 2 | 0.024652 | 0.024652 | 0.012326 | 26.2 | 0 |
| Transverse speed (mm/min) | 2 | 0.026496 | 0.026496 | 0.013248 | 28.17 | 0 |
| Stand of Distance (mm) | 2 | 0.116652 | 0.116652 | 0.058326 | 124 | 0 |
| Error | 20 | 0.009407 | 0.009407 | 0.00047 | | |
| Total | 26 | 0.177207 | | | | |

R-Square value = 94.69%.

standoff distance increases the kerf width. A similar trend was attained for a combination of transverse speed and higher standoff distance. An increase in transverse speed results in a high traverse rate that results in less overlapping of composite surface machining and lesser imposing over machining surface; this fact reduces kerf width. From the combination plot, it is also inferred that lower standoff distance and pump pressure alone with higher transverse speed is the optimal machining condition for machining the 0.3 wt.% reduced graphene oxide loaded composite.

## Conclusion

Reduced graphene oxide and MMT nanofiller added polymer composite with glass fibre reinforcement are successfully fabricated through compression moulding, and its machinability behaviour is analysed through AWJM with kerf width as the response variable. The conclusion derived from the experimentation is as follows

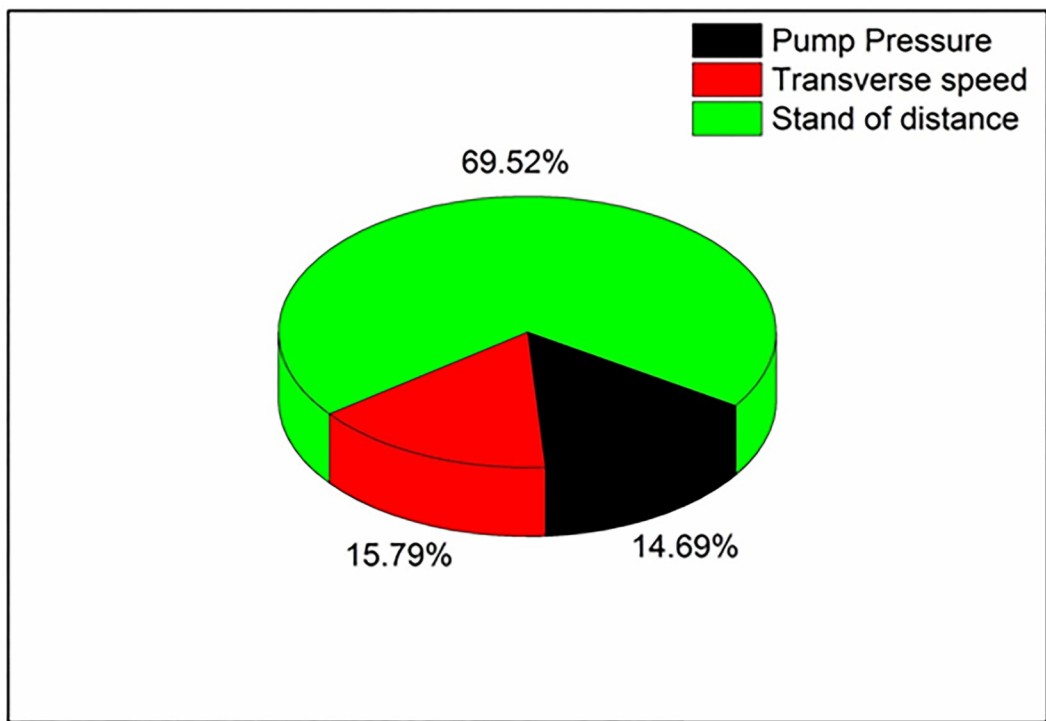

**Fig 10. Contribution of machining parameters in 0.2 wt. % filler loaded composite.**

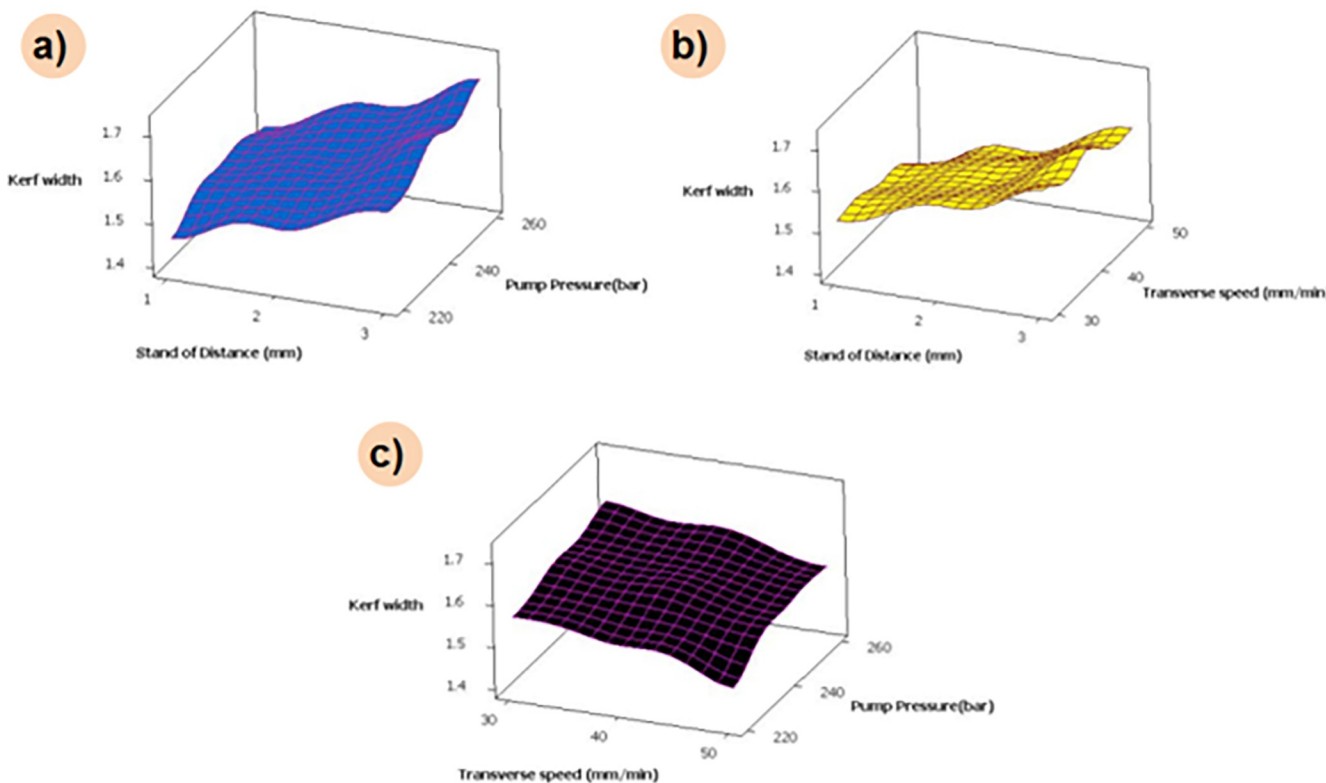

**Fig 11. Influence of AWJM parameters on kerf width for 0.2 wt. % of filler loaded composite.**

Table 9. Response table for 0.3 wt. % of filler loaded composite.

| Level | Pump Pressure (bar) | Transverse speed (mm/min) | Stand of Distance (mm) |
|-------|---------------------|---------------------------|------------------------|
| 1 | 1.634 | 1.711 | 1.594 |
| 2 | 1.681 | 1.676 | 1.66 |
| 3 | 1.707 | 1.636 | 1.768 |
| Delta | 0.072 | 0.076 | 0.173 |
| Rank | 3 | 2 | 1 |

Table 10. ANOVA for 0.3 wt. % of reduced graphene oxide loaded composite.

| Source | DF | Seq SS | Adj SS | Adj MS | F—Test | P-value |
|--------|----|--------|--------|--------|--------|---------|
| **Pump pressure(bar)** | 2 | 0.024141 | 0.024141 | 0.01207 | 22.11 | 0 |
| **Transverse speed (mm/min)** | 2 | 0.025719 | 0.025719 | 0.012859 | 23.55 | 0 |
| **Stand of Distance (mm)** | 2 | 0.137874 | 0.137874 | 0.068937 | 126.28 | 0 |
| **Error** | 20 | 0.010919 | 0.010919 | 0.000546 | | |
| **Total** | 26 | 0.198652 | | | | |

R-Square value = 94.50%.

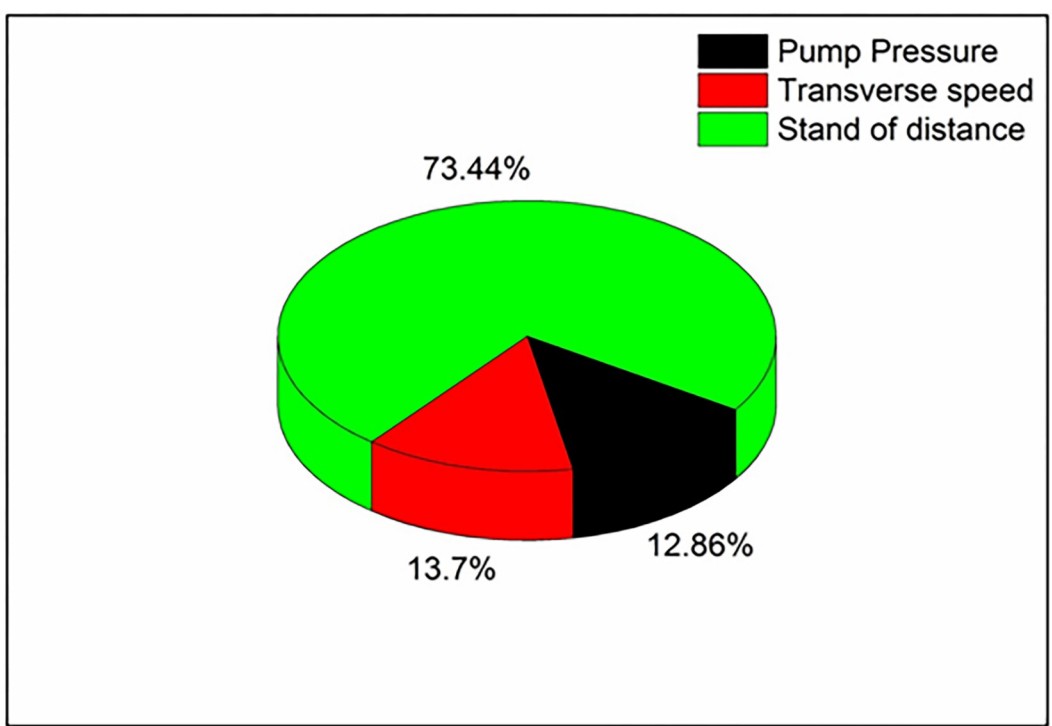

**Fig 12. Contribution of machining parameters in 0.1 wt. % filler loaded composite.**

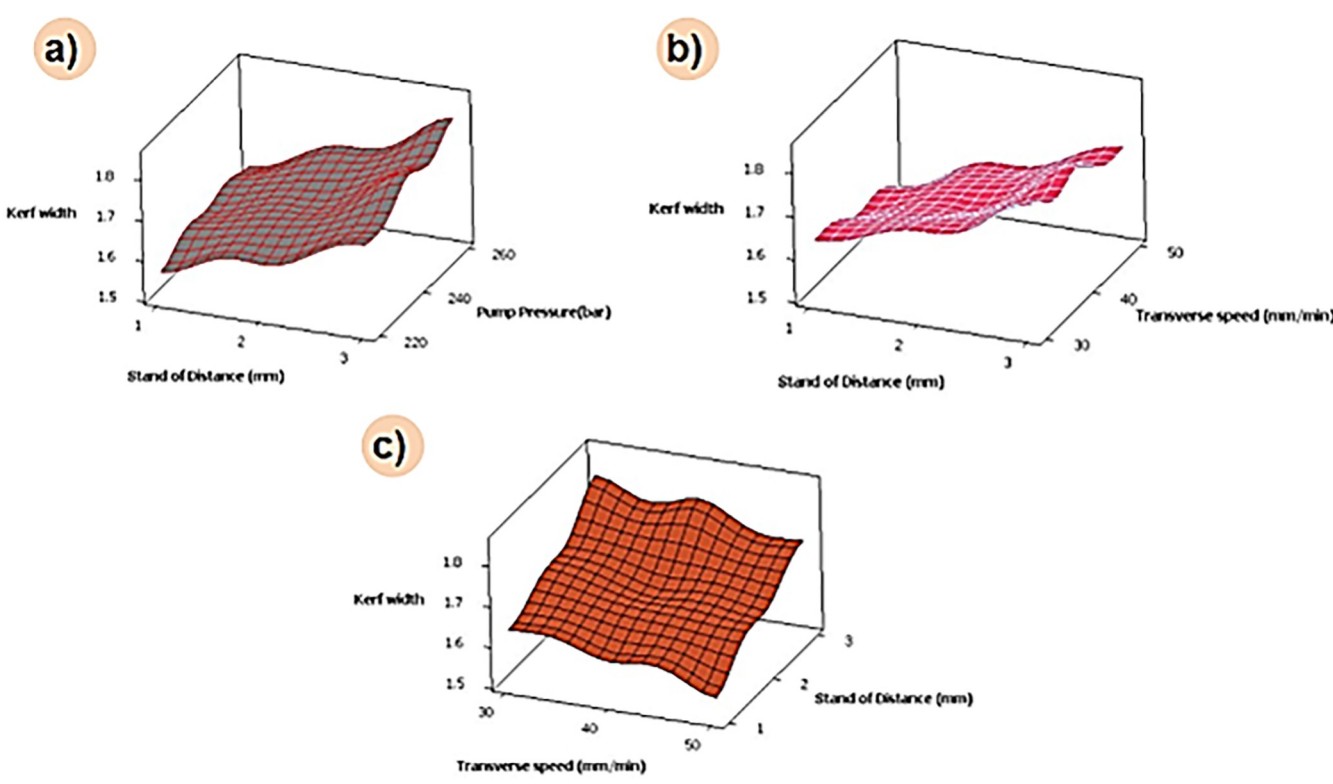

**Fig 13. Influence of AWJM parameters on kerf width for 0.3 wt. % of filler loaded composite.**

- The addition of rGO filler has a significant effect over kerf width, which decreases with the addition of rGO up to 0.2% and kerf width increases for further addition of rGO.

- Analysis of Variance results exposes that stand of distance has a major effect over kerf width irrespective of rGO %.

- An increase in standoff distance and pump pressure increases the kerf width, whereas the kerf width is low at a higher transverse speed of the nozzle.

- The influence of pump pressure over kerf width decreases slightly with an increase in rGO filler addition.

## Supporting information

**S1 File.**
(DOCX)

## Author Contributions

**Conceptualization:** Kavimani V., Gopal P. M., Stalin B., Balasubramani V., Dhinakaran V., Nagaprasad N., Leta Tesfaye Jule, Krishnaraj Ramaswamy.

**Data curation:** Kavimani V., Gopal P. M., Stalin B., Balasubramani V., Dhinakaran V., Nagaprasad N., Leta Tesfaye Jule, Krishnaraj Ramaswamy.

**Formal analysis:** Kavimani V., Gopal P. M., Stalin B., Balasubramani V., Dhinakaran V., Nagaprasad N., Leta Tesfaye Jule, Krishnaraj Ramaswamy.

**Investigation:** Gopal P. M., Stalin B., Balasubramani V., Dhinakaran V., Nagaprasad N., Leta Tesfaye Jule, Krishnaraj Ramaswamy.

**Methodology:** Balasubramani V.

**Resources:** Kavimani V., Dhinakaran V., Nagaprasad N., Leta Tesfaye Jule, Krishnaraj Ramaswamy.

**Software:** Kavimani V.

**Supervision:** Kavimani V., Gopal P. M., Stalin B., Dhinakaran V., Nagaprasad N., Leta Tesfaye Jule, Krishnaraj Ramaswamy.

**Validation:** Kavimani V., Gopal P. M., Stalin B., Balasubramani V., Dhinakaran V., Nagaprasad N., Leta Tesfaye Jule, Krishnaraj Ramaswamy.

**Visualization:** Kavimani V., Gopal P. M., Stalin B., Balasubramani V., Dhinakaran V., Nagaprasad N., Leta Tesfaye Jule, Krishnaraj Ramaswamy.

**Writing – original draft:** Kavimani V., Gopal P. M., Nagaprasad N., Leta Tesfaye Jule, Krishnaraj Ramaswamy.

**Writing – review & editing:** Stalin B.

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
