## [Decision Letter · Decision Letter 0]

15 Apr 2022

PONE-D-22-08465Influence of reduced graphene oxide addition on kerf width in abrasive water jet machining of nano filler added Epoxy-Glass Fibre compositePLOS ONE

Dear Dr. Ramaswamy,

Thank you for submitting your manuscript to PLOS ONE. After careful consideration, we feel that it has merit but does not fully meet PLOS ONE’s publication criteria as it currently stands. Therefore, we invite you to submit a revised version of the manuscript that addresses the points raised during the review process.

We look forward to receiving your revised manuscript.

Kind regards,

Yasir Nawab, PhD

Academic Editor

PLOS ONE

Journal Requirements:

“The funders had no role in study design, data collection and analysis, decision to publish, or preparation of the manuscript”

5. Please upload a new copy of Figure 8 as the detail is not clear. Please follow the link for more information: https://blogs.plos.org/plos/2019/06/looking-good-tips-for-creating-your-plos-figures-graphics/

6. Please ensure that you refer to Figure 9 in your text as, if accepted, production will need this reference to link the reader to the figure.

7. We note you have included a table to which you do not refer in the text of your manuscript. Please ensure that you refer to Table 3 in your text; if accepted, production will need this reference to link the reader to the Table.

Reviewers' comments:

Reviewer's Responses to Questions

**Comments to the Author**

1. Is the manuscript technically sound, and do the data support the conclusions?

Reviewer #1: Partly

Reviewer #2: Yes

2. Has the statistical analysis been performed appropriately and rigorously? 

Reviewer #1: Yes

Reviewer #2: Yes

3. Have the authors made all data underlying the findings in their manuscript fully available?

Reviewer #1: No

Reviewer #2: Yes

4. Is the manuscript presented in an intelligible fashion and written in standard English?

Reviewer #1: No

Reviewer #2: Yes

5. Review Comments to the Author

Reviewer #1: Comments and Suggestions for Authors

The manuscript “Influence of reduced graphene oxide addition on kerf width in abrasive water jet machining of nano filler added Epoxy-Glass Fibre composite” by Krishnaraj et.al. present novel and interesting results that deserve to be published after several minor improvements:

1. It will be of great interest if the Surface roughness at varied Water Jet Pressure and material removal rate at different traverse speed should be included in detail with proper interpretation of results.

2. Scanning electron microscope is the most important testing technique to understand the distribution of nanoparticles as well the surface information. The results are missing, and it should be included in it.

3. Mechanical testing in combinations for detecting flexural, impact, tensile, density and hardness properties should be included as well.

4. Rheological properties are important in order to understand the nanofiller and epoxy blends. Please included these properties.

5. The Figure captions are much unsatisfactory. It is a fundamental responsibility of the authors to make Figures self-explanatory. Go to the guidelines and examples and prepare the Figure captions accordingly. This applies to all Figures with this manuscript.

6. FTIR data should be included.

Reviewer #2: Authors need to provide minor information and include in the paper as mentioned below.

1) Provide the L27 experiments combinations.

2) Provide the measured readings of the kerf in a table with the instrument used for the measurement of kerf.

3) Figure 2 and Figure 3 should be explained in detail with their relevance.

4) Fabricated composites pictures to be provided.

5) Effect on the composites layers can be discussed during the cutting.

6. PLOS authors have the option to publish the peer review history of their article (what does this mean?). If published, this will include your full peer review and any attached files.

Reviewer #1: No

Reviewer #2: No

---

## [Author Response · Author response to Decision Letter 0]

28 May 2022

Response to reviewer:

Reviewer 1

Comment 1: It will be of great interest if the Surface roughness at varied Water Jet Pressure and material removal rate at different traverse speed should be included in detail with proper interpretation of results.

Response: Thanks for your valuable comments. This article is mainly focus on effect of machining parameters on kerf width here the other output parameter are neglected. The parameter such as surface roughness and material removal rate are optimized and interpreted as separate manuscript.

Comment 2: Scanning electron microscope is the most important testing technique to understand the distribution of nanoparticles as well the surface information. The results are missing, and it should be included in it.

Response: presence of filler has been confirmed by XRD analysis it is already published in our article .https://www.hindawi.com/journals/apt/2021/6627743/

Comment 3: Mechanical testing in combinations for detecting flexural, impact, tensile, density and hardness properties should be included as well.

Response: Thanks for your valuable suggestion. The mechanical properties of the developed r-GO-MMT reinforced composite have been already published in our article. Please refer the link .https://www.hindawi.com/journals/apt/2021/6627743/.The necessary citation has been made in this revised manuscript 

These developed composite exhibits better mechanical and flame retardancy properties upto 0.3 wt. % addition of r-GO fillers. The detailed investigation on mechanical behaviour of developed composite was illustrated in our previous report [29]. 

Comment 4: Rheological properties are important in order to understand the nanofiller and epoxy blends. Please include these properties.

Response: This manuscript is extensively focus on machinability of developed composite hence rheological studies may be neglected 

Comment 5: The Figure captions are much unsatisfactory. It is a fundamental responsibility of the authors to make Figures self-explanatory. Go to the guidelines and examples and prepare the Figure captions accordingly. This applies to all Figures with this manuscript.

Response: As per reviewer recommendation captions of the figure has been modified 

Comment 6 : FTIR data should be included.

Response: Fundamental characterization has been made in our previous manuscript and the necessary citation has been made.

Reviewer 2

Comments 1) provide the L27 experiments combinations.

Response: As per reviewer suggestion L27 OA has been added in this revised manuscript.

A B C Kerf width (µm) S/N ratio

 0% 0.1 % 0.2% 0.3% 0% 0.1 % 0.2% 0.3% 

220 30 1 1.46 1.41 1.50 1.61 -3.29 -2.98 -3.52 -4.14

220 30 2 1.51 1.49 1.56 1.67 -3.58 -3.46 -3.86 -4.45

220 30 3 1.56 1.54 1.64 1.75 -3.86 -3.75 -4.30 -4.86

220 40 1 1.43 1.39 1.48 1.57 -3.11 -2.86 -3.41 -3.92

220 40 2 1.49 1.48 1.54 1.61 -3.46 -3.41 -3.75 -4.14

220 40 3 1.56 1.52 1.59 1.72 -3.86 -3.64 -4.03 -4.71

220 50 1 1.37 1.35 1.40 1.52 -2.73 -2.61 -2.92 -3.64

220 50 2 1.41 1.40 1.46 1.59 -2.98 -2.92 -3.29 -4.03

220 50 3 1.52 1.48 1.57 1.67 -3.64 -3.41 -3.92 -4.45

240 30 1 1.52 1.51 1.55 1.67 -3.64 -3.58 -3.81 -4.45

240 30 2 1.55 1.52 1.59 1.67 -3.81 -3.64 -4.03 -4.45

240 30 3 1.64 1.61 1.67 1.80 -4.30 -4.14 -4.45 -5.11

240 40 1 1.49 1.46 1.53 1.58 -3.46 -3.29 -3.69 -3.97

240 40 2 1.51 1.48 1.55 1.66 -3.58 -3.41 -3.81 -4.40

240 40 3 1.63 1.60 1.69 1.79 -4.24 -4.08 -4.56 -5.06

240 50 1 1.43 1.39 1.47 1.57 -3.11 -2.86 -3.35 -3.92

240 50 2 1.49 1.47 1.53 1.64 -3.46 -3.35 -3.69 -4.30

240 50 3 1.59 1.56 1.64 1.75 -4.03 -3.86 -4.30 -4.86

260 30 1 1.49 1.46 1.52 1.63 -3.46 -3.29 -3.64 -4.24

260 30 2 1.62 1.57 1.65 1.75 -4.19 -3.92 -4.35 -4.86

260 30 3 1.66 1.64 1.72 1.85 -4.40 -4.30 -4.71 -5.34

260 40 1 1.49 1.45 1.52 1.64 -3.46 -3.23 -3.64 -4.30

260 40 2 1.56 1.52 1.58 1.67 -3.86 -3.64 -3.97 -4.45

260 40 3 1.66 1.60 1.73 1.84 -4.40 -4.08 -4.76 -5.30

260 50 1 1.44 1.40 1.47 1.56 -3.17 -2.92 -3.35 -3.86

260 50 2 1.53 1.49 1.56 1.68 -3.69 -3.46 -3.86 -4.51

260 50 3 1.61 1.58 1.63 1.74 -4.14 -3.97 -4.24 -4.81

Comments 2) Provide the measured readings of the kerf in a table with the instrument used for the measurement of kerf.

Response: As per reviewer console, the kerf width measurement has been detailed in this revised manuscript.

Kerf width is taken as the output parameter and it is measured by calculating the distance between machined slots using Leica microscope coupled with image analyser as depicted in figure 2b.

Comments 3) Figure 2 and Figure 3 should be explained in detail with their relevance.

Response: Necessary correction has been made as per reviewer recommendation

Scatter-plot matrix is used to identify the correlation among the set of variable pairs. These pair wise correlations can be organizes in a form of matrix. In general present matrix pair in diagonal order depicts the better correlation among the matrix pairs and absents of outliers in the attained output data’s. Hierarchical cluster is an analysis worked by algorithm in which the similar data’s are grouped in a form of clusters. Herein each clusters are varied from each other and the values are widely similar to each other. From figure 4 it can be noted that the values of output parameter are grouped into four clusters with two expressive colour blue and brown (dark and lighter).These are order at the range of -1 to 1 with respect to input parameter. The values of output parameters are predominant ordering from top to bottom displayed in the form of lighter and dark colours of blue and brown.

4) Fabricated composites pictures to be provided.

Response: as per reviewer recommendation pictures of fabricated composite has been added

Comments 5) Effect on the composites layers can be discussed during the cutting.

Response: This work is mainly focus on influence of machining parameter on kerf width hence effect of composite layers are neglected in this manuscript

---

## [Decision Letter · Decision Letter 1]

12 Jun 2022

Influence of reduced graphene oxide addition on kerf width in abrasive water jet machining of nanofiller added Epoxy-Glass Fibre composite

PONE-D-22-08465R1

Dear Dr. Ramaswamy,

We’re pleased to inform you that your manuscript has been judged scientifically suitable for publication and will be formally accepted for publication once it meets all outstanding technical requirements.

Kind regards,

Yasir Nawab, PhD

Academic Editor

PLOS ONE

Additional Editor Comments (optional):

Reviewers' comments:

Reviewer's Responses to Questions

**Comments to the Author**

1. If the authors have adequately addressed your comments raised in a previous round of review and you feel that this manuscript is now acceptable for publication, you may indicate that here to bypass the “Comments to the Author” section, enter your conflict of interest statement in the “Confidential to Editor” section, and submit your "Accept" recommendation.

Reviewer #1: All comments have been addressed

Reviewer #2: All comments have been addressed

2. Is the manuscript technically sound, and do the data support the conclusions?

Reviewer #1: Yes

Reviewer #2: Yes

3. Has the statistical analysis been performed appropriately and rigorously? 

Reviewer #1: N/A

Reviewer #2: Yes

4. Have the authors made all data underlying the findings in their manuscript fully available?

Reviewer #1: Yes

Reviewer #2: Yes

5. Is the manuscript presented in an intelligible fashion and written in standard English?

Reviewer #1: Yes

Reviewer #2: Yes

6. Review Comments to the Author

Reviewer #1: (No Response)

Reviewer #2: (No Response)

7. PLOS authors have the option to publish the peer review history of their article (what does this mean?). If published, this will include your full peer review and any attached files.

Reviewer #1: No

Reviewer #2: No

---

## [Editor Report · Acceptance letter]

8 Aug 2022

PONE-D-22-08465R1 

Influence of reduced graphene oxide addition on kerf width in abrasive water jet machining of nanofiller added Epoxy-Glass Fibre composite 

Dear Dr. Ramaswamy:

I'm pleased to inform you that your manuscript has been deemed suitable for publication in PLOS ONE. Congratulations! Your manuscript is now with our production department. 

Kind regards, 

on behalf of

Dr. Yasir Nawab 

Academic Editor

PLOS ONE